# The Effect of Endometriosis on In Vitro Fertilization Outcomes: A Systematic Review and Meta-Analysis

**DOI:** 10.3390/healthcare12232435

**Published:** 2024-12-03

**Authors:** Ilenia Mappa, Zoe Pauline Page, Daniele Di Mascio, Chiara Patelli, Francesco D’Antonio, Antonella Giancotti, Francesco Gebbia, Giulia Mariani, Mauro Cozzolino, Ludovico Muzii, Giuseppe Rizzo

**Affiliations:** 1Department of Obstetrics and Gynecology, Fondazione Policlinico di Tor Vergata, University of Rome Tor Vergata, 00133 Roma, Italy; ilenia.mappa@ospedalecristore.it (I.M.); zoe.page@uniroma2.it (Z.P.P.); 2Department of Maternal and Child Health and Urological Sciences, Sapienza University of Rome, 00185 Roma, Italy; daniele.dimascio@uniroma1.it (D.D.M.); antonella.giancotti@uniroma1.it (A.G.); ludovico.muzii@uniroma1.it (L.M.); 3Department of Obstetrics and Gynecology, University of Verona, 37129 Verona, Italy; chiara.patelli@univr.it; 4Department of Obstetrics and Gynecology, University of Chieti, 66013 Chieti, Italy; francesco.dantonio@unich.it; 5IVIRMA Global Research Alliance, IVI Roma-Rome, 00197 Roma, Italy; francesco.gebbia@ivirma.com (F.G.); giulia.mariani@ivirma.com (G.M.); mauro.cozzolino@ivirma.com (M.C.)

**Keywords:** endometriosis, IVF, clinical pregnancy rate, live birth rate, fertilization rate, implantation rate, meta-analysis

## Abstract

Objectives: The purpose of this study was to evaluate the impact of endometriosis on various outcomes of in vitro fertilization (IVF), including live birth rates, clinical pregnancy rates, fertilization rates, and implantation rates, through a systematic review and meta-analysis. Methods: Systematic searches were carried out using PubMed, MEDLINE, Cochrane Central Register of Controlled Trials, Scopus, EMBASE, and Web of Science from January 2010 to November 2023. Studies comparing IVF outcomes in women with and without endometriosis were included. The primary outcome was live birth rate; secondary outcomes included clinical pregnancy, fertilization, and implantation rates. Data were extracted and analyzed using odds ratio (OR) and 95% confidence interval (CI) with fixed or random-effects models, depending on heterogeneity. Results: From 1340 studies initially identified, 40 studies met the inclusion criteria, encompassing 8970 women with endometriosis and 42,946 control participants. There were no significant differences between the endometriosis and control groups in terms of live birth rate (OR 1.03, 95% CI 0.75–1.41, *p* = 0.84), clinical pregnancy rate (OR 0.86, 95% CI 0.72–1.02, *p* = 0.1), or fertilization rate (OR 0.96, 95% CI 0.79–1.15, *p* = 0.64). However, endometriosis was associated with a significantly lower implantation rate (OR 0.85, 95% CI 0.74–0.97, *p* = 0.02). Conclusions: Endometriosis significantly negatively affects implantation rates in women undergoing IVF, despite the absence of significant differences in live birth, clinical pregnancy, and fertilization rates. Further research is needed to evaluate the impact of different stages of endometriosis on IVF outcomes and to develop optimized management protocols for these patients.

## 1. Introduction

Endometriosis is an inflammatory disorder characterized by the presence of endometrial tissue outside of the uterine cavity [1]. Women with endometriosis may experience a range of gynecological symptoms, including pelvic pain, dysmenorrhea, dyspareunia, and fertility issues [2].

It is estimated to affect about 10% of women of reproductive age. However, it may reach 25–40% among infertile women [3].

The high prevalence of endometriosis in the infertile population suggests a complex and multi-factorial role in infertility, which may affect ovarian reserve, ovulation, tubal anatomy, embryo quality, implantation, and pregnancy development [4]. As a consequence, the impact of endometriosis on embryo implantation is specifically difficult to highlight, and the exact mechanisms connecting endometriosis and infertility are still unclear and incompletely understood. Indeed the etiology of endometriosis remains incompletely elucidated to date. Its physiopathology is also debated. Its high prevalence in the infertile population (20–30%) underlines its very complex and multi-factorial role in infertility, affecting ovarian reserve, ovulation, tubal anatomy, embryo quality, implantation, and pregnancy development. Thus, the impact of endometriosis on embryo implantation is specifically difficult to highlight. While the relationship between infertility and severe/advanced stage endometriosis may be explained by the severe anatomical and functional abnormalities and lower ovarian reserve, the cause of infertility in the case of minimal or mild endometriosis is still unclear and remains a matter of debate [5].

The principal underlying cause of endometriosis related-infertility is most likely chronic pelvic inflammation. This inflammation can lead to increased levels of cytokines and macrophages in the peritoneal fluid of patients with endometriosis and has been linked to adverse effects on human reproduction [6,7,8,9]. Chronic inflammation can result in the formation of pelvic adhesions and scarring, distorting pelvic anatomy and leading to structural changes that may affect the normal functioning of reproductive organs. These adhesions might obstruct or alter the fallopian tubes, disrupt ovarian function, or hinder the release, movement, or transport of oocytes, thus impeding fertilization [10]. Other mechanisms that may explain reduced fertility include hormonal imbalances [11], uterine hyper- and dysperistalsis [12], and follicular loss along with intraovarian vascular injury due to ovarian stromal fibrosis [13,14]. Further endometriosis may induce immunological or hormonal abnormalities that by themselves may affect IVF outcomes [15,16].

IVF may be useful in bypassing some of the negative effects of endometriosis; however, women with endometriosis may still encounter infertility treatments [17].

Moreover, it is logical to predict that some of these causes cannot be avoided by using IVF, and there is an uncertainty if women affected by endometriosis may respond worse to IVF as compared to patients with other infertility causes. There is also a lack of agreement on how the severity of endometriosis affects IVF outcome and underscores the importance of investigating the relationship between endometriosis and IVF outcome.

Our objective was to quantify the impact of endometriosis on live birth, clinical pregnancy, fertilization, and implantation rates through a comprehensive meta-analysis, and the meta-analysis aimed to provide a thorough and evidence-based overview of how endometriosis affects various aspects of IVF, including live birth rates, pregnancy rates, and other relevant outcomes. This information can be valuable for clinicians, researchers, and patients in understanding the potential impact of endometriosis on fertility treatment outcomes and guiding clinical decision-making.

## 2. Materials and Methods

### 2.1. Search Strategy and Selection Criteria

Systematic electronic database searches were carried out using PubMed, MEDLINE, Cochrane Central Register of Controlled Trials, Scopus, EMBASE, and Web of Science. The following keywords were searched for: fertilization in vitro, fertilization in vitro/methods, ART, ICSI, IVF, endometriosis, endometriosis/complications, endometriosis/pathology, endometrioma, DIE, outcome/s, pregnancy, pregnancy rate, and birth rate. Boolean operators including endometriosis and all the IVF outcomes were also searched. Only articles in the English language were included. The search was performed between January 2010 and November 2023. This study protocol is registered in the Prospero database (2023 CRD42023467215).

The population group of this study included women who had undergone or were undergoing IVF procedures; the study group was made up of women affected by endometriosis diagnosed either by ultrasound, MRI, or histologic examination, whereas the control group consisted of women without endometriosis.

The following articles were excluded: studies that had included women that had undergone treatment using donor oocytes or that had received surgical or medical treatment for endometriosis prior to ART; studies that did not provide full texts or lacked necessary data for analysis; studies without a control group; case reports or abstracts; studies limited to conference proceedings; and non-original or duplicate publications.

The primary outcome was live birth rate (LBR), which was determined as the number of deliveries resulting in a live newborn.

The secondary outcomes were as follows:(a)Clinical pregnancy rate (CPR), defined by the ultrasound visualization of an intrauterine gestational sac;(b)Fertilization rate (FR), defined as the percentage of transformation of micro injected oocytes into two pronuclei;(c)Implantation rate (IR): number of implanted embryos/number of embryos transferred.
CPR and LBR data were expressed as total number of ART patients considered in the study, and when this was not possible, we considered the data per transfer or per cycle.

### 2.2. Data Extraction and Quality Assessment

Two authors (IM and CP) reviewed all abstracts independently and selected the articles that fit the predefined selection criteria. Subsequently, the authors reviewed the full manuscripts of these selected publications to determine final inclusion or exclusion decisions. Any disagreements regarding the relevance of articles were settled through consensus or, if necessary, by consulting another reviewer (GR). Disagreement among authors was present only for 2 studies, which were re-reviewed with GR. The quality of the included studies was evaluated using the Newcastle–Ottawa scale, where each study is evaluated based on the selection of the study groups, the comparability of these groups, and the outcome of interests.

### 2.3. Statistical Analysis

An analysis of the ART outcomes of women with endometriosis compared to a control group was performed. Data were extracted from original studies; if raw data were not provided, we used the available data for calculations. From these indices, we derived corresponding odds ratios (ORs) and their 95% confidence intervals. Meta-analysis of Observational Studies in Epidemiology (MOOSE) guidelines were observed in conducting this study. ORs from individual studies were combined using either a fixed-effects model or a random-effects model. The latter analysis was adopted when a high heterogeneity was found. Heterogeneity was assessed using I^2^ statistics and visually with forest plots. High heterogeneity was considered when I^2^ exceeded 50%. STATA version 13.1 (Stata Corp., College Station, TX, USA) was used for all statistical analyses.

## 3. Results

From the initial search, we identified 1340 studies (Figure 1, Prisma chart and Prisma checklist Appendix A). We conducted an initial screening of all titles and abstracts to evaluate their relevance to our study based on our predefined inclusion criteria. This screening process led to the exclusion of 1152 publications.

Another set of 148 publications were excluded following a review of the entire texts of the remaining studies: (60 studies published before 2010, 15 without any relevant outcome measures, 7 concerning adenomyosis, 5 because data were not extractable from the studies, 8 were a review or meta-analysis, 9 had no control group or intra-patient control groups, 2 were about performing IVF with egg donation, 2 were not in the English language, and 40 were about patients that had undergone surgical or medical treatment before IVF (Appendix A).

Consequently, the systematic review included a total of 40 studies [17,18,19,20,21,22,23,24,25,26,27,28,29,30,31,32,33,34,35,36,37,38,39,40,41,42,43,44,45,46,47,48,49,50,51,52,53,54,55,56] (Table 1).

The Newcastle–Ottawa Quality Assessment Scale scores for these studies are presented in Table 2. We analyzed data from 8970 women diagnosed with endometriosis and 42,946 control participants.

In Table 3, a summary of the results obtained is presented.

### 3.1. Live Births

Information on live birth rates was available from 26 studies. In 15 of these studies, live birth rates were reported per total number of women undergoing IVF (study group n. 3210/8186 vs. control group n 7124/30,247). No significant difference in live birth rates between the endometriosis and control group was found (OR 1.03, 95% CI = 0.75–1.41, *p* = 0.84) (Figure 2). A high heterogeneity was observed among these studies, with an I^2^ value of 93.5% (95% CI = 91.3% to 94.9%).

When combining results from four studies that evaluated live birth per transfer (study group n. 274/748 vs. control group n 942/2483), no difference in live birth rate was observed between the two groups (OR 0.93, 95% CI = 0.78–1.10, *p* = 0.388) (Appendix A). The I^2^ value was 0% (95% CI = 0% to 67.9%).

When pooling results from seven studies that reported live birth per cycle (study group n. 398/435 vs. control group n. 270/7462), no significant difference was observed in live birth rate (OR 0.99, 95% CI = 0.86–1.15, *p* = 0.89) (Appendix A). The I^2^ value was 0% (95% CI = 0% to 58.5%), suggesting low heterogeneity among the studies.

### 3.2. Clinical Pregnancies

Data on the clinical pregnancy rate were available in 30 studies. In 23 of these, pregnancy rate was expressed per total number of women undergoing IVF (study group n. 1833/6552 vs. control group n. 9716/34,691). After combining the data, no difference in clinical pregnancy rates was found between the endometriosis and control groups (OR 0.86, 95% CI = 0.72–1.02, *p* = 0.098) (Figure 3). A high heterogeneity was observed, with an I^2^ value of 72.5% (95% CI = 56.2% to 80.9%).

When pooling data from four studies that evaluated clinical pregnancy per transfer (study group n. 345/759 vs. control group n 996/1856), no difference in clinical pregnancy rates was observed (OR 0.91, 95% CI = 0.76–1.09, *p* = 0.29) (Appendix A). The I^2^ value was 0% (95% CI = 0% to 67.9%), indicating minimal variability among the studies.

Combining data from three studies that described clinical pregnancy per cycle as an outcome (study group n. 271/990 vs. control group n 505/1571) also showed no statistically significant difference (OR 0.95, 95% CI = 0.76–1.17, *p* = 0.61) (Appendix A). The I^2^ value was 0% (95% CI = 0% to 72.9%), indicating a low heterogeneity.

### 3.3. Fertilization Rate

Fertilization rate data, defined as the number of fertilized oocytes per number of metaphase II oocytes, were extracted from eight studies (study group: 4153 out of 5661; control group: 43,979 out of 62,972). No difference in fertilization rates was found (OR = 0.96, 95% CI = 0.79–1.15, *p* = 0.64) (Figure 4). A significant heterogeneity was indicated by an I^2^ value of 80.5% (95% CI = 57.6% to 88.5%).

### 3.4. Implantation Rate

Data on implantation rates, defined as successful implantations per total number of embryos transferred, were available in 10 studies (study group: 406 out of 2032; control group: 3573 out of 11,120). Heterogeneity was limited, with an I^2^ value of 37.5% (95% CI = 0% to 68.9%) (Figure 5). Pooling the results from these ten studies showed a statistically significant reduction in implantation rates (OR = 0.85, 95% CI = 0.74–0.97, *p* = 0.02).

## 4. Discussion

### 4.1. Main Findings

This systematic review and meta-analysis indicate that endometriosis among patients undergoing IVF/ICSI, regardless of its stage, does not significantly impact live birth, clinical pregnancy, and fertilization rates when compared to other causes of infertility. However, a notable reduction in implantation rate was observed in women with endometriosis.

### 4.2. Strengths and Limitations

One major strength of this study is its restriction of our analysis to studies published from 2010 onwards, minimizing the influence of outdated IVF techniques. Furthermore, we selected only studies in which women with endometriosis had not undergone any previous surgical or medical treatments for this condition. We adopted the Newcastle–Ottawa Quality Assessment Scale to ensure that high-quality studies with a low bias were included. As a consequence, the inclusion of recent research and the strict selection criteria enhanced the quality of the analysis.

However, the main limitation was the lack of differentiation between different grades of endometriosis, potentially increasing heterogeneity. Similarly, it was not possible to stratify women according to their characteristics (e.g., age, BMI) and IVF protocols that similarly may increase heterogenicity. Nevertheless, clinical heterogeneity is common in meta-analyses and does not necessarily affect generalizability more than individual studies do [58]. Other limitations concerned data presentation variability as some studies did not report the number of patients included but only the number of cycles, where some studies reported IVF results after a certain number of cycles while others only followed one cycle. Moreover, the diagnosis of endometriosis was not performed the same way in all studies (either by laparoscopy or ultrasound). This may affect the correct diagnosis and classification of endometriosis since deep endometriosis lesions may be missed with ultrasound in the absence of a high level of expertise of sonographers. Similarly, we cannot exclude the small form of endometriosis in the control group due to the limitations of the non-invasive technique in their diagnosis.

Further, the concomitant association with adenomyosis, a potential risk factor of decreased implantation, was not evaluated in the studies considered [59]. Similarly, the association with other causes of infertility (such as both male and female factors) that may affect IVF outcome independently by endometriosis could not be extracted by the studies considered. Finally, it cannot be ignored that the patients of some studies might have received medical or surgical therapy without being reported.

### 4.3. Interpretation

In this systematic review and meta-analysis, which included 40 studies, we compared 8970 infertile women with endometriosis and 42,946 controls with other causes of infertility. Our findings suggest that endometriosis at all stages does not impact major fertility outcomes such as live birth, clinical pregnancy, and fertilization rates but is associated with a decreased implantation rate. These results align with the findings of Qu et al. [60], who also found a significant difference in implantation rate among women undergoing IVF/ICSI with or without endometriosis. They also demonstrated that the endometriosis and control group had similar pregnancy, live birth, and fertilization rates. Unlike our study, their exclusion criteria were less stringent (the single criterion of exclusion was egg donation), potentially weakening their findings. To mitigate this, we carefully selected studies that excluded patients who had undergone any previous medical or surgical interventions.

The reduction in the implantation rate of women with endometriosis is of interest. This finding may have a triple etiology: reduced oocyte and embryo quality, endometrial receptivity defect, and an altered interaction between the embryo and the endometrium. Our analysis did not allow us to differentiate the role of these potential factors, and it is not possible to disregard that this may play a different role in each woman.

A limitation of our study is the lack of differentiation by disease stage. Harb et al. [60] found over a decade ago that severe endometriosis (stages III/IV) is associated with a reduction in implantation and clinical pregnancy rates, with a trend towards reduced live births that did not reach statistical significance, whereas mild endometriosis (stage I/II) did not seem to affect these outcomes. However, their study included publications from 1985, and IVF techniques have evolved since then, which is why we included only studies from 2010 onwards.

Our study shows that endometriosis is associated with a reduced implantation rate. Successful implantation involves coordinated interactions between a receptive endometrium and a fertilized oocyte, suggesting multiple causal links between endometriosis and reduced implantation rates [61]. Abnormal inflammatory factors in women with endometriosis can lead to a hormonal imbalance, affecting estradiol secretion and progesterone resistance [10], which, in turn, may modify endometrial receptivity. Additionally, ectopic endometrial tissue in these women shows significant biochemical and ultrastructural differences from normal tissue [62]. Further, women with endometriosis may have a lower implantation rate due to oocyte and consequent embryo quality or different [63]. These findings could have clinical implications and justify the development of personalized pre-implantation protocols for women with endometriosis scheduled for IVF. Future research agenda will include the search for a specific genetic profile of women with endometriosis and a transcriptomic analysis of the endometrium leading to targeted treatments on the individual mechanism inducing endometriosis-related infertility.

## Figures and Tables

**Figure 1 healthcare-12-02435-f001:**
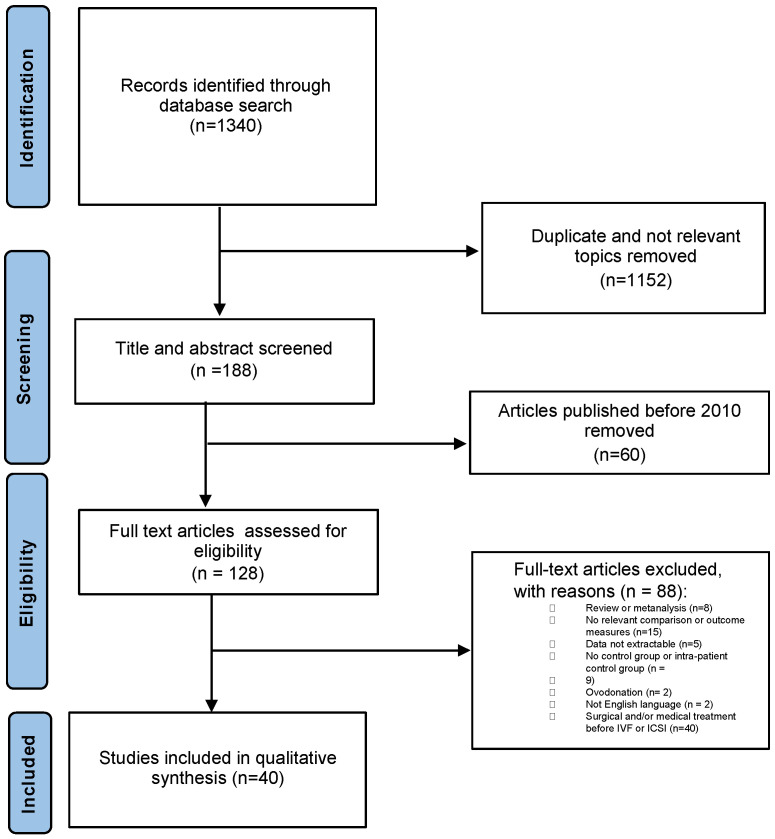
Prisma chart showing study selection for systematic review on the effect of endometriosis on IVF treatment outcome.

**Figure 2 healthcare-12-02435-f002:**
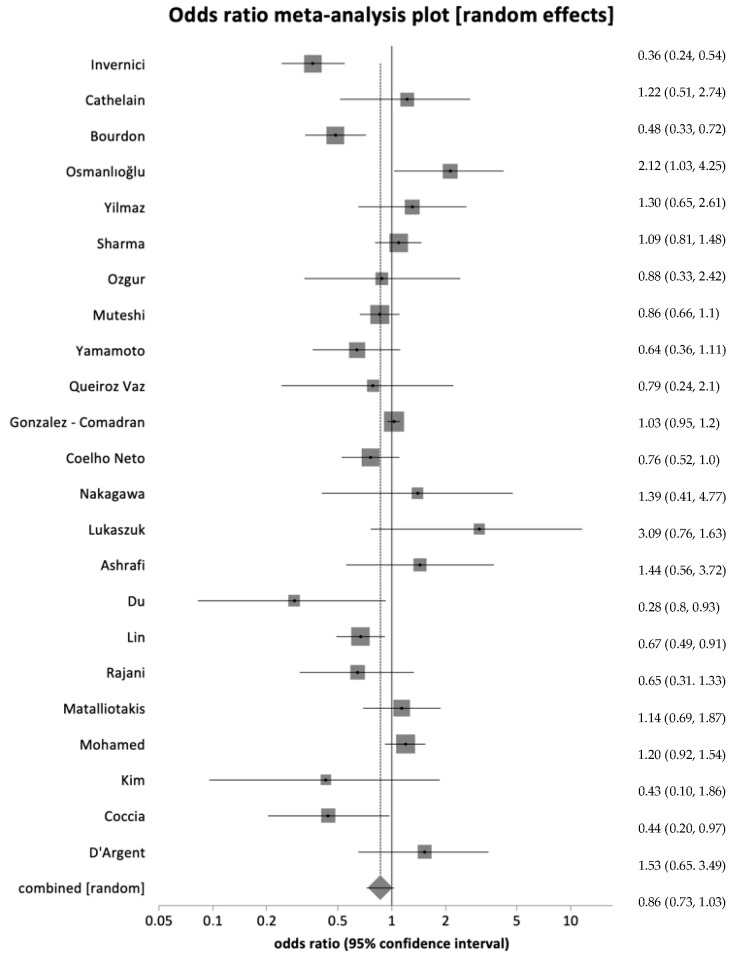
Forest plot showing live birth rate per total number of women with no significant difference in live birth rates across groups [17,19,20,23,28,31,32,33,34,36,37,38,41,45,46,47,50,51,52,53,54,55,56].

**Figure 3 healthcare-12-02435-f003:**
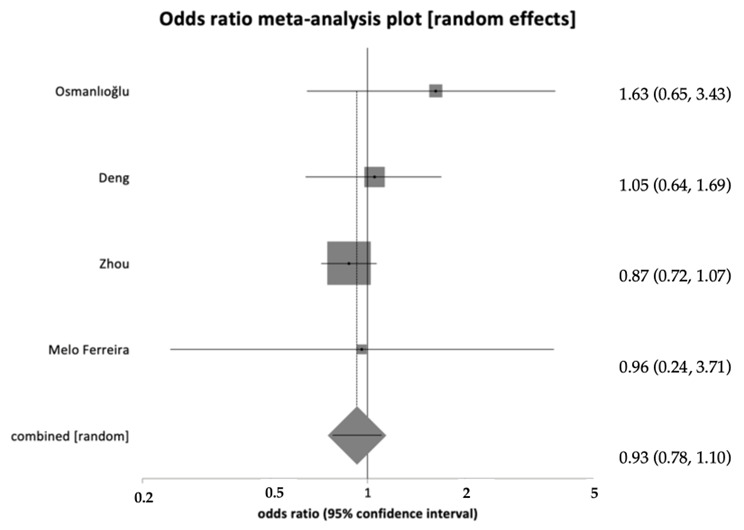
Forest plot showing clinical pregnancy per total number of women with no significant difference in live birth rates across groups [23,24,25,29].

**Figure 4 healthcare-12-02435-f004:**
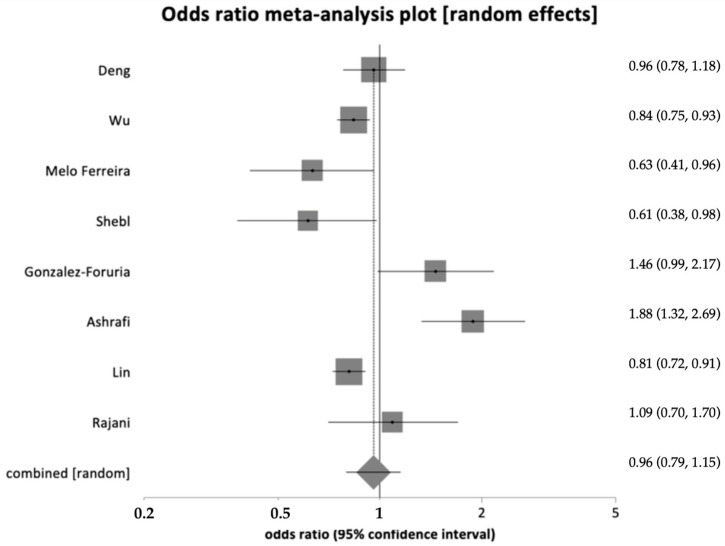
Forest plot showing fertilization rate with no significant difference in live birth rates across groups [24,26,30,35,36,46,50,51].

**Figure 5 healthcare-12-02435-f005:**
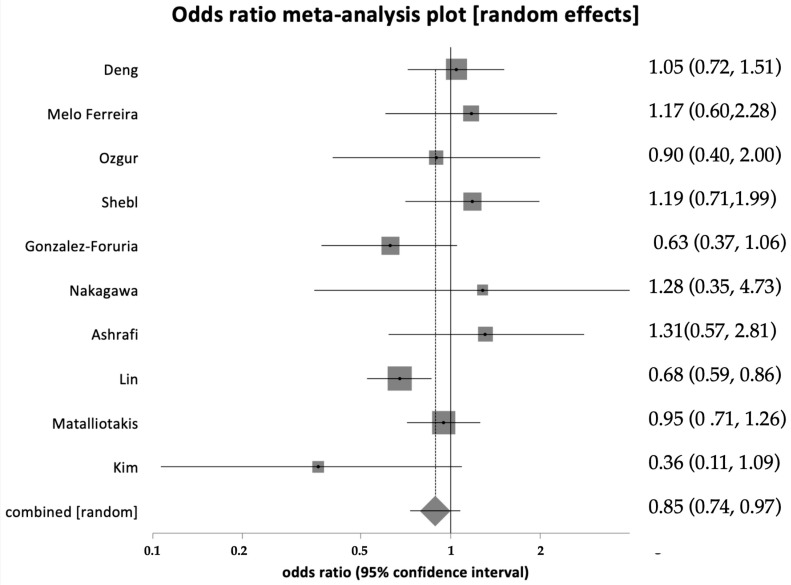
Forest plot showing implantation rate significantly lower in women with endometriosis [24,30,32,35,36,41,46,50,52].

**Table 1 healthcare-12-02435-t001:** General characteristics of the included studies.

Author	Year	Country	Period	Endometriosis Group	Control Group	Outcomes	Study Design
Invernici [17]	2022	Italy	Jan 2014–Dec 2020.	Endometriosis (n = 218)	No endometriosis (n = 248)	CPR, LBR	Retrospective study
Chen [18]	2023	China	Feb 2019–Dec 2020	Endometriosis (n = 412)	No endometriosis (n = 1551)	IR, CPR, LBR	Retrospective study
Cathelain [19]	2023	France	Nov 2019–Jun 2021	Endometriosis (n = 73)	No endometriosis (n = 205)	CPR	Prospective cohort study
Bourdon [20]	2023	France	Sep 2016–Sep 2018	Endometriosis (n = 190)	No endometriosis (OHSS risk) (n = 281)	CPR, LBR	Prospective cohort study
Li [21]	2023	China	May 2021–Apr 2022	Ovarian endometriosis (n = 37)	No endometriosis (n = 37)	FR, IR, CPR	Prospective study
Wang [22]	2023	Taiwan	Jan 2016–Dec 2019	Endometriosis (n = 229) cycles	No endometriosis/ademyosis (n = 1002) cycles	CPR, LBR	Retrospective study
Osmanlıoğlu [23]	2022	Turkey	Jan 2015–Dec 2020	Ovarian endometriosis (n = 64)	No endometrioma (n = 225)	CPR, LBR	Retrospective case control study
Deng [24]	2022	China	Jan 2014–Dec 2018	Ovarian endometriosis (n = 90) 191 cycles	No endometriosis (n = 403) 888 cycles	FR, IR, LBR	Retrospective case control study
Zhou [25]	2022	China	Jan 2012–Dec 2014	Endometriosis (n = 433)	Tubal factor infertility (n = 1299)	FR, CPR, LBR	Retrospective study
Wu [26]	2021	China	Jan 2011–Dec 2019	Ovarian endometriosis (n = 293 without surgery)	No endometriosis (n = 862)	FR, CPR, LBR	Retrospective study
Boucret [27]	2020	France	Jan 2014–Mar 2018	Endometriosis (n = 90) 155 cycles	No endometriosis (n = 590) 969 cycles	LBR	Retrospective cohort study
Yilmaz [28]	2019	Turkey	Mar 2015–Mar 2018	Ovarian endometriosis (n = 73), unilateral or bilateral	No endometriosis (n = 86)	CPR, LBR	Retrospective study
Feichtinger [29]	2019	Sweden	Jan 2009–Dec 2013	Endometriosis (n = 172)	No endometriosis (n = 2585)	LBR	Cross-sectional case control
Melo Ferreira [30]	2019	Brazil	Oct 2009–Oct 2010	Endometriosis (n = 35)	Male factor infertility (n = 60)	FR, IR, CPR LBR	Prospective case control study
Sharma [31]	2019	India	Jan 2010–Jan 2015	Endometriosis (n = 355).	Tubal factor infertility (n = 466)	CPR, LBR	Retrospective cohort study
Ozgur [32]	2018	Turkey	Sep2014–Sep 2016	Ovarian endometirosis (n = 29), unilateral or bilateral	No endometrioma (n = 60)	IR, CPR	Retrospective matched case control study
Muteshi [33]	2018	UK	Jan 2000–Dec 2014	Endometriosis (n = 531), all stages	No endometriosis (n = 737)	CPR, LBR	Retrospective cohort study
Yamamoto [34]	2017	USA	Jan 2008–Dec 2009	Endometriosis (n = 68)	No endometriosis (n = 649)	CPR	Retrospective study
Shebl [35]	2017	Austria	2013	Endometriosis (n = 114) 129 cycles	No endometriosis (n = 119) 129 cycles	FR, IR, LBR	Retrospective case control study
Gonzalez [36]–Comadran [37]	2017	Spain	Jan 2010–Dec 2012	Endometriosis (n = 3583)	No endometriosis (n = 18,833)	FR, CPR, LBR	Retrospective cohort study
Queiroz Vaz [38]	2017	Brazil	Jan 2007–Dec 2013	DIE (n = 27)	No endometriosis (n = 154)	CPR	Retrospective cohort study
Coelho Neto [39]	2016	Brazil	Jan 2011–Dec 2012	Endometriosis (n = 241)	No endometriosis (n = 546)	CPR, LBR	Retrospective cohort study
Senapati [40]	2016	USA	2008–2010	Endometriosis (n = 12,335) cycles	Tubal factor infertility (n = 22,778), unexplained (n = 38,713), others (n = 196,295)–No endometriosis 257,786	IR, LBR	Retrospective cohort study
Gonzalez-Foruria [41]	2016	Spain	Jan 2010–Dec 2014	Endometriosis (n = 101) 326 cycles	Male factor infertility (n = 68) 202 cycles; Tubal factor infertility (n = 44) 125 cycles; Unexplained infertility (n = 81) 243 cycles; Mixed aetiology (n = 26) 51 cycles–No endometriosis 219	FR, IR, CPR	Retrospective cohort study
Nakagawa [42]	2015	Japan	Dec 2011–Jul 2013	Ovarian endometirosis (n = 26), unilateral	Male factor and unexplained infertility (n = 29)	IR, CPR	Retrospective case control study
Coelho Neto [43]	2015	Brazil	Jan 2011–Dec 2012	Endometrioma (n = 39)	No endometrioma (n = 478)	CPR	Retrospective cohort study
Borges [44]	2015	Brazil	Jan 2005–May 2014	Endometriosis (n = 431) cycles stages III/IV	No endometriosis (n =) 2510 cycles	FR, CPR	Prospective cohort study
Polat [45]	2014	Turkey	Jul 2005–Nov 2012	Endometriosis (n = 485)	Tubal factor infertility (n = 131)	CPR, LBR	Retrospective case control study
Lukaszuk [46]	2014	Poland	May 2007–Jan 2011	Moderate-to-severe endometriosis (n = 23) (49 cycles)	No endometriosis (n = 78)	CPR, LBR	Retrospective study
Ashrafi [47]	2014	Iran	Mar 2005–Dec 2007	Ovarian endometriosis (n = 47)	Mild male factor infertility (n = 57)	FR, IR, CPR	Prospective cohort study
Du [48]	2013	Japan	2010	Endometriosis (n = 29)	No endometriosis (n = 36)	CPR	Retrospective study
Benaglia [49]	2013	Spain, Italy	Jan 2006–Jul 2010	Ovarian endometriosis (n = 39)	No endometriosis (n = 78)	IR, CPR, LBR	Retrospective cohort study
Benaglia [50]	2012	Italy	Jan 2005–Dec 2009	Ovarian endometriosis (n = 78)	No endometriosis (n = 156)	LBR	Retrospective cohort study
Lin [51]	2012	China	Jan 2006–Dec 2010	Endometriosis (n = 177)	No endometriosis (n = 4267)	FR, IR, CPR	Retrospective cohort study
Rajani [52]	2012	India	nr	Endometriosis (n = 56)	No endometriosis (n = 111)	FR, CPR	Prospective study
Matalliotakis [53]	2011	USA	1996–2002	Endometriosis (n = 130)	No endometriosis (n = 163)	IR, CPR, LBR	Retrospective cohort study
Mohamed [54]	2011	UK	Jan 2000–Dec 2008	Endometriosis (n = 415) 415 cycles	No endometriosis (n = 6871) 6871 cycles	CPR, LBR	Retrospective cohort study
Kim [55]	2011	Korea		Endometriosis (n = 20) (20 cycles)	Tubal factor infertility (n = 20) 20 cycles	IR, CPR, LBR	Retrospective cohort study
Coccia [56]	2011	Italy	Feb 2001–Mar 2007	Endometriosis (n = 148)	Tubal factor infertility (n = 72)	CPR	Retrospective cohort study
D’Argent [57]	2010	France	Jan 2007–Oct 2008	Colorectal endometriosis (n = 29)	No endometriosis (n = 497)	CPR, LBR	Retrospective study

**Table 2 healthcare-12-02435-t002:** Newcastle–Ottawa Quality Assessment Scale of the included studies.

Author	Case Cohort Representative	Selection of Non-Exposed Cohort	Ascertaiment of Exposure	Outcome Negative at Start	Comparability by Design	Comparability by Analysis	Outcome Assessment	Duration of Follow Up	Score
Invernici [17]	*	*	*	*	*	*	*	*	8
Chen [18]	*	*	*	*	*	*	*	*	8
Cathelain [19]	*	*	*	*	*	*	*	*	8
Bourdon [20]	*	*	*	*	*	*	*	x	7
Li [21]	*	*	*	*	*	*	*	*	8
Wang [22]	*	*	*	*	*	*	*	*	8
Osmanlıoğlu [23]	*	*	*	*	*	*	*	*	8
Deng [24]	*	*	*	*	*	*	*	*	8
Zhou [25]	*	*	*	*	*	*	*	*	8
Wu [26]	*	*	*	*	*	*	*	x	7
Boucret [27]	*	*	*	*	*	*	*	x	7
Yilmaz [28]	*	x	*	*	*	*	*	x	6
Feichtinger [29]	*	*	*	x	*	*	*	*	7
Melo Ferreira [30]	*	*	*	*	*	*	*	*	8
Sharma [31]	*	*	*	*	*	*	*	*	8
Ozgur [32]	*	*	*	*	*	*	x	*	7
Muteshi [33]	*	*	*	*	*	*	*	*	8
Yamamoto [34]	*	*	*	*	*	x	*	*	7
Shebl [35]	*	*	*	*	*	*	*	*	8
Gonzalez [36]–Comadran [37]	*	*	*	*	*	*	*	*	8
Queiroz Vaz [38]	*	*	*	*	*	*	*	*	8
Coelho Neto [39]	*	*	*	*	**	*	*	*	9
Senapati [40]	*	*	*	*	*	*	*	+	8
Gonzalez-Foruria [41]	*	*	*	*	**	*	*	*	8
Nakagawa [42]	*	*	*	*	*	*	*	*	8
Coelho Neto [43]	*	*	*	*	*	*	*	x	7
Borges [44]	*	*	*	*	**	*	*	*	9
Polat [45]	*	*	*	x	*	*	*	x	6
Lukaszuk [46]	*	*	*	*	*	*	*	*	8
Ashrafi [47]	*	*	*	*	*	*	*	*	8
Du [48]	*	*	*	*	*	*	*	*	8
Benaglia [49]	*	*	*	*	**	*	*	*	9
Benaglia [50]	*	*	*	*	**	*	*	*	9
Lin [51]	*	*	*	*	*	*	*	*	8
Rajani [52]	*	*	*	*	*	*	*	*	8
Matalliotakis [53]	*	*	*	*	*	*	*	*	8
Mohamed [54]	*	*	*	*	*	*	x	*	7
Kim [55]	*	*	*	x	*	*	x	*	6
Coccia [56]	*	*	*	*	*	*	*	x	7
D’Argent [57]	*	*	*	*	**	*	*	*	9

* indicates that the feature is present; x indicates that the feature is absent.

**Table 3 healthcare-12-02435-t003:** Data analysis summary.

Outcome	Studies	Population (Case vs. Control)	Pooled OR (95% CI)	I^2^ (%)	*p*-Values
Livebirth (per person)	15	3210/8186 vs. 7124/30,247	1.03 (0.75–1.41)	93.5	0.845
Livebirth (per transfer)	4	274/748 vs. 942/2483	0.93 (0.78–1.10)	0	0.388
Livebirth (per cycle)	7	398/435 vs. 270/7462	0.99 (0.86–1.15)	0	0.890

Clinical pregnancy (per person)	23	1833/6552 vs. 9716/34,691	0.86 (0.72–1.02)	72.5	0.098
Clinical pregnancy (per transfer)	4	345/759 vs. 996/1856	0.91 (0.76–1.09)	0	0.292
Clinical pregnancy (per cycle)	3	271/990 vs. 505/1571	0.95 (0.76–1.17)	0	0.616

Fertilization rate	8	4153/5661 vs. 43,979/62,972	0.96 (0.79–1.15)	80.5	0.641

Implantation rate	10	406/2032 vs. 3573/11,120	0.85 (0.74–0.97)	37.5	0.023

## Data Availability

Available from the authors at reasonable request.

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
