# Peer review of "The Effect of Endometriosis on In Vitro Fertilization Outcomes: A Systematic Review and Meta-Analysis"

_healthcare, 2024, doi:10.3390/healthcare12232435_

Round 1

Reviewer 1 Report

Comments and Suggestions for Authors

1. This article aimed to illustrate the impact of endometriosis on IVF outcomes. The background section described the causes of endometriosis-related infertility, some of which cannot be bypassed by IVF. Consequently, these factors potentially influence IVF outcomes and represent the principal challenges in treating endometriosis-related infertility through IVF, which can be briefly described. 

2. The results in the forest plots do not fully align with those presented in Table 3.

3. The discussion section lacks sufficient elaboration, and the results are not adequately explained.

4. Line 178, the number of decimal places in the p-values in Table 3 is not uniform.

Author Response

1 This article aimed to illustrate the impact of endometriosis on IVF outcomes. The background section described the causes of endometriosis-related infertility, some of which cannot be bypassed by IVF. Consequently, these factors potentially influence IVF outcomes and represent the principal challenges in treating endometriosis-related infertility through IVF, which can be briefly described. 

Thank you for your suggestion.  This important message is now reported in the introduction section

  1. The results in the forest plots do not fully align with those presented in Table 3.

We moved table 3 at the beginning of the results to allow a more easier interpretation and check for inconsistences

  1. The discussion section lacks sufficient elaboration, and the results are not adequately explained.

Thanks for the suggestion we improved the discussion

  1. Line 178, the number of decimal places in the p-values in Table 3 is not uniform.

We apologize. Now corrected

Reviewer 2 Report

Comments and Suggestions for Authors

Esteemed Authors and Editorial team,

I think the theme behind the study is a very valid one considering the burden of endometriosis in infertility patients nowadays. It goes without saying that we cannot have enough interest and research concerning this association.

The study under evaluation is well designed, well conducted and well written.

The authors have also correctly identified most limitations. I would add to this chapter that perhaps a correct diagnosis and classification of endometriosis cannot be warranted in most studies since it was done either by laparoscopy or ultrasound (perhaps missing deep endometriosis lesions - laparoscopy, or perhaps not always diagnosing the entire extent of disease if not endometriosis ultrasound experts). Since diagnosis and classification is not yet standardized, this is a shortcoming which ca not easily be overcome.

Otherwise, I support the publication of the article in the current form.

Author Response

Review 2

I think the theme behind the study is a very valid one considering the burden of endometriosis in infertility patients nowadays. It goes without saying that we cannot have enough interest and research concerning this association.

The study under evaluation is well designed, well conducted and well written.

The authors have also correctly identified most limitations. I would add to this chapter that perhaps a correct diagnosis and classification of endometriosis cannot be warranted in most studies since it was done either by laparoscopy or ultrasound (perhaps missing deep endometriosis lesions - laparoscopy, or perhaps not always diagnosing the entire extent of disease if not endometriosis ultrasound experts). Since diagnosis and classification is not yet standardized, this is a shortcoming which ca not easily be overcome.

Otherwise, I support the publication of the article in the current form.

Thanks for your positive comments

We fully agree with your comment now added in the discussion

Reviewer 3 Report

Comments and Suggestions for Authors

I would like to thank you for sharing this very interesting work and to applaud your efforts in preparing this paper. In my opinion, the article needs some supplementary work to meet the journal's standards.

Abstract: Ok.
Keywords: Ok.

Introduction: Although there are several areas where it could be improved, the manuscript's introduction provides a solid starting point for discussing endometriosis and its impact on infertility.

  • The significance of researching the precise effects of endometriosis on various IVF outcomes (such as live birth rate and clinical pregnancy rate) could be explained more precisely in the introduction. For example, the authors could provide a brief explanation of current research or theories on how endometriosis might impact IVF success.
  • Although mechanisms like pelvic inflammation and structural alterations are mentioned in the introduction, it would be helpful to delve further into these mechanisms or present studies that have explored these relationships. To define the study's focus, specific references to immunological or hormonal abnormalities that affect IVF outcomes could be added.
  • Include a sentence or two outlining the gaps in the current research, especially the lack of agreement on how the severity of endometriosis affects IVF outcomes. Highlighting this gap could support the study’s aim and underscore its significance.
  • The objective is briefly mentioned, but it could be stated more explicitly. For example, instead of “evaluate the impact of endometriosis on various outcomes of IVF,” a more focused statement such as “to quantify the impact of endometriosis on live birth, clinical pregnancy, fertilization, and implantation rates through a comprehensive meta-analysis” would be clearer and more specific.

Materials and Methods: To increase transparency and reproducibility, the Materials and Methods section could benefit from additional details.

  • Specific search terms and Boolean operators used in each database, such as "endometriosis AND IVF outcomes," could be added to the search strategy. This would ensure that no important studies were missed and allow other researchers to replicate the search exactly.
  • Although some exclusion criteria are listed, it would be beneficial to be more specific. Since mixed infertility factors (such as both male and female factors) can influence IVF outcomes independently of endometriosis, it would help to specify if studies involving mixed infertility factors were included. This would help readers better understand the study selection process.
  • Provide a more thorough definition of each IVF-related outcome, such as clinical pregnancy rate, fertilization rate, and live birth rate (e.g., whether outcomes are per transfer, per cycle, or per patient). Describe how studies with different reporting formats (e.g., per cycle vs. per patient) were handled. Clearly defining these terms would enhance the meta-analysis’s validity, as different criteria can lead to inconsistent findings.
  • Include more details on the quality assessment process. For example, specify how any disagreements regarding data extraction or study selection were resolved (e.g., through discussion or consultation with a third reviewer). This would increase confidence in the accuracy and reliability of the data used.

Results: Although the Results section is well-structured, the following suggestions could make it clearer and ensure the results are presented in an understandable manner:

  • Clarify the consistency of outcome reporting by describing how each measure (e.g., live birth rate, clinical pregnancy rate) was handled across studies. For instance, specify when rates are provided per cycle, per transfer, or per patient, and ensure this distinction is consistently applied. Consistently stating how IVF outcomes are measured would improve comparability and reduce reader confusion.
  • Consider presenting subgroup analyses based on endometriosis severity or other factors (e.g., patient age, IVF protocols) to further explore heterogeneity. If this is not feasible, mention it as a limitation in the findings. This would allow for a more nuanced interpretation of the results and might indicate whether endometriosis impacts IVF success more for certain subgroups.
  • Include brief explanations for each forest plot, such as "implantation rate significantly lower in endometriosis group" or "no significant difference in live birth rates across groups." While the forest plots are helpful, adding textual interpretations of the main findings would make it easier for readers to follow.
  • Consider including a summary table of the main findings, with p-values, heterogeneity (I²) values, pooled odds ratios, and confidence intervals for each major outcome (e.g., live birth rate, clinical pregnancy rate, implantation rate). A summary table would improve readability and provide a quick reference for readers.

Discussion: Although the Discussion section provides a good overview, there are areas where it could be expanded or refined to strengthen the interpretation of the findings.

  • A more thorough comparison with previous studies and meta-analyses would enhance the discussion. This might include addressing any discrepancies in results, particularly in studies linking endometriosis to other IVF outcomes (e.g., clinical pregnancy or live birth rates). Placing the findings in the context of existing literature could help readers understand how this study contributes to the field and why its findings might differ from previous studies.
  • It would be beneficial to provide more specific recommendations for future research, such as analyzing the influence of endometriosis stage, evaluating the significance of different IVF protocols, or examining long-term outcomes in patients with endometriosis. Specific research recommendations could help guide future work and strengthen the study’s contribution to understanding endometriosis and IVF.
  • Consider adding a dedicated section on the strengths of this study, which would enhance the credibility of the findings (e.g., only recent research included, strict selection criteria). Acknowledging both the strengths and limitations of the study would offer a balanced perspective and highlight its contributions.

References: Although the article's bibliography seems comprehensive, it could benefit from a few additional references, especially where the material would be strengthened by more recent or foundational studies.

  • Adding references that describe the mechanisms by which endometriosis affects fertility, such as immune system dysregulation, hormonal imbalances, and endometrial receptivity, could strengthen the introduction and discussion. These sources would provide a stronger theoretical foundation for the study and give readers a more complete understanding of how endometriosis might impact fertility.

Author Response

I would like to thank you for sharing this very interesting work and to applaud your efforts in preparing this paper. In my opinion, the article needs some supplementary work to meet the journal's standards.

Abstract: Ok.
Keywords: Ok.

Introduction: Although there are several areas where it could be improved, the manuscript's introduction provides a solid starting point for discussing endometriosis and its impact on infertility.

thanks

  • The significance of researching the precise effects of endometriosis on various IVF outcomes (such as live birth rate and clinical pregnancy rate) could be explained more precisely in the introduction. For example, the authors could provide a brief explanation of current research or theories on how endometriosis might impact IVF success.

Thanks for the suggestion done.

  • Although mechanisms like pelvic inflammation and structural alterations are mentioned in the introduction, it would be helpful to delve further into these mechanisms or present studies that have explored these relationships. To define the study's focus, specific references to immunological or hormonal abnormalities that affect IVF outcomes could be added.

Done new ref 15 and 15

  • Include a sentence or two outlining the gaps in the current research, especially the lack of agreement on how the severity of endometriosis affects IVF outcomes. Highlighting this gap could support the study’s aim and underscore its significance.

Excellent suggestion done

  • The objective is briefly mentioned, but it could be stated more explicitly. For example, instead of “evaluate the impact of endometriosis on various outcomes of IVF,” a more focused statement such as “to quantify the impact of endometriosis on live birth, clinical pregnancy, fertilization, and implantation rates through a comprehensive meta-analysis” would be clearer and more specific.

We agree thanks done as suggested

Materials and Methods: To increase transparency and reproducibility, the Materials and Methods section could benefit from additional details.

  • Specific search terms and Boolean operators used in each database, such as "endometriosis AND IVF outcomes," could be added to the search strategy. This would ensure that no important studies were missed and allow other researchers to replicate the search exactly.

done

  • Although some exclusion criteria are listed, it would be beneficial to be more specific. Since mixed infertility factors (such as both male and female factors) can influence IVF outcomes independently of endometriosis, it would help to specify if studies involving mixed infertility factors were included. This would help readers better understand the study selection process.

We agree on this point but it was not reported in the studies considered. We acknowledged in the discussion as a limitation

  • Provide a more thorough definition of each IVF-related outcome, such as clinical pregnancy rate, fertilization rate, and live birth rate (e.g., whether outcomes are per transfer, per cycle, or per patient). Describe how studies with different reporting formats (e.g., per cycle vs. per patient) were handled. Clearly defining these terms would enhance the meta-analysis’s validity, as different criteria can lead to inconsistent findings.

Done as also suggested by reviewer 1

  • Include more details on the quality assessment process. For example, specify how any disagreements regarding data extraction or study selection were resolved (e.g., through discussion or consultation with a third reviewer). This would increase confidence in the accuracy and reliability of the data used.

done

Results: Although the Results section is well-structured, the following suggestions could make it clearer and ensure the results are presented in an understandable manner:

  • Clarify the consistency of outcome reporting by describing how each measure (e.g., live birth rate, clinical pregnancy rate) was handled across studies. For instance, specify when rates are provided per cycle, per transfer, or per patient, and ensure this distinction is consistently applied. Consistently stating how IVF outcomes are measured would improve comparability and reduce reader confusion.

We simplify the results section

  • Consider presenting subgroup analyses based on endometriosis severity or other factors (e.g., patient age, IVF protocols) to further explore heterogeneity. If this is not feasible, mention it as a limitation in the findings. This would allow for a more nuanced interpretation of the results and might indicate whether endometriosis impacts IVF success more for certain subgroups.

Unfortunately it  was not possible and we mention as a limitation

  • Include brief explanations for each forest plot, such as "implantation rate significantly lower in endometriosis group" or "no significant difference in live birth rates across groups." While the forest plots are helpful, adding textual interpretations of the main findings would make it easier for readers to follow.

done

  • Consider including a summary table of the main findings, with p-values, heterogeneity (I²) values, pooled odds ratios, and confidence intervals for each major outcome (e.g., live birth rate, clinical pregnancy rate, implantation rate). A summary table would improve readability and provide a quick reference for readers.

Already present Table 3

Discussion: Although the Discussion section provides a good overview, there are areas where it could be expanded or refined to strengthen the interpretation of the findings.

  • A more thorough comparison with previous studies and meta-analyses would enhance the discussion. This might include addressing any discrepancies in results, particularly in studies linking endometriosis to other IVF outcomes (e.g., clinical pregnancy or live birth rates). Placing the findings in the context of existing literature could help readers understand how this study contributes to the field and why its findings might differ from previous studies.

Done as suggested also by others reviewers

  • It would be beneficial to provide more specific recommendations for future research, such as analyzing the influence of endometriosis stage, evaluating the significance of different IVF protocols, or examining long-term outcomes in patients with endometriosis. Specific research recommendations could help guide future work and strengthen the study’s contribution to understanding endometriosis and IVF.

Thanks added as also suggested by the academic editor a future research agenda

  • Consider adding a dedicated section on the strengths of this study, which would enhance the credibility of the findings (e.g., only recent research included, strict selection criteria). Acknowledging both the strengths and limitations of the study would offer a balanced perspective and highlight its contributions.

done

References: Although the article's bibliography seems comprehensive, it could benefit from a few additional references, especially where the material would be strengthened by more recent or foundational studies.

  • Adding references that describe the mechanisms by which endometriosis affects fertility, such as immune system dysregulation, hormonal imbalances, and endometrial receptivity, could strengthen the introduction and discussion. These sources would provide a stronger theoretical foundation for the study and give readers a more complete understanding of how endometriosis might impact fertility.

New references added

Reviewer 4 Report

Comments and Suggestions for Authors

The article is a correctly conducted meta-analysis. The main limitations of the work are the heterogeneity of the disease and methods of its diagnosis, as well as methods for assessing the outcomes of ART programs. The authors point to this in the Discussion section. There are two small remarks:

1. There is a mistake in table 3 in description of "clinical pregnancy" groups. It should be "per person/per transfer/per sycle.

2. Another limitation of the study is the selection of the control group, since ultrasound and MRI, which have been identified as the main diagnostic methods, cannot exclude small forms of endometriosis

Author Response

The article is a correctly conducted meta-analysis. The main limitations of the work are the heterogeneity of the disease and methods of its diagnosis, as well as methods for assessing the outcomes of ART programs. The authors point to this in the Discussion section.

Thanks for the positive comments

 There are two small remarks:

  1. There is a mistake in table 3 in description of "clinical pregnancy" groups. It should be "per person/per transfer/per sycle.

We apologize corrected

  1. Another limitation of the study is the selection of the control group, since ultrasound and MRI, which have been identified as the main diagnostic methods, cannot exclude small forms of endometriosis

We agree this limitation was added as also suggested by reviewer 2

Round 2

Reviewer 3 Report

Comments and Suggestions for Authors

Dear Authors, your consistent efforts to improve the manuscript in my opinion made it eligible for publication.

Author Response

Dear Authors, your consistent efforts to improve the manuscript in my opinion made it eligible for publication.

Thanks for your positive comments